# A Review of the Emerging White Chick Hatchery Disease

**DOI:** 10.3390/v13122435

**Published:** 2021-12-04

**Authors:** Kerry McIlwaine, Christopher J. Law, Ken Lemon, Irene R. Grant, Victoria J. Smyth

**Affiliations:** 1School of Biological Sciences, Queen’s University, Belfast BT9 5DL, UK; kmcilwaine04@qub.ac.uk (K.M.); c.law@qub.ac.uk (C.J.L.); i.grant@qub.ac.uk (I.R.G.); 2Virology Branch, Veterinary Sciences Division, Agri-Food and Biosciences Institute, Belfast BT4 3SD, UK; kenneth.lemon@afbini.gov.uk

**Keywords:** white chick, hatchery disease, chicken astrovirus, vertical transmission

## Abstract

White chick hatchery disease is an emerging disease of broiler chicks with which the virus, chicken astrovirus, has been associated. Adult birds typically show no obvious clinical signs of infection, although some broiler breeder flocks have experienced slight egg drops. Substantial decreases in hatching are experienced over a two-week period, with an increase in mid-to-late embryo deaths, chicks too weak to hatch and pale, runted chicks with high mortality. Chicken astrovirus is an enteric virus, and strains are typically transmitted horizontally within flocks via the faecal–oral route; however, dead-in-shell embryos and weak, pale hatchlings indicate vertical transmission of the strains associated with white chick hatchery disease. Hatch levels are typically restored after two weeks when seroconversion of the hens to chicken astrovirus has occurred. Currently, there are no commercial vaccines available for the virus; therefore, the only means of protection is by good levels of biosecurity. This review aims to outline the current understanding regarding white chick hatchery disease in broiler chick flocks suffering from severe early mortality and increased embryo death in countries worldwide.

## 1. Introduction

Hatchery diseases occur when eggs become infected by pathogens, often leading to drops in the numbers of eggs laid, reduced hatches and/or embryonic defects such as dwarfing. True vertical transmission occurs when the pathogen is passed from a parent into the egg while pseudo or apparent vertical transmission arises from contaminated faeces on the eggshell, allowing pathogens to penetrate the egg, impacting embryonic development. There are several pathogens known to be passed to the developing embryo from parent birds, including *Salmonella pullorum* and *S*. *gallinarum,* certain *Mycoplasma* species, avian leucosis virus, avian encephalomyelitis virus (AEV), fowl adenovirus and chicken astrovirus (CAstV) [1,2,3], resulting in high levels of chick mortality and impairment in chicks after hatching. Typical of vertically transmitted viruses such as AEV and CAstV, the parent flock will rapidly seroconvert, providing neutralising antibodies to prevent vertical virus transmission and deposit maternally-derived antibodies into the eggs to protect developing embryos and neonates.

White chick hatchery disease is a recently emerged disease affecting broiler chicks and embryos that has been reported in a number of regions worldwide. This disease results in severe weakness in chicks that either prevents hatching, or chicks survive only briefly post-hatch with characteristic white down [4]. Although it had been problematic in North America for several decades [5], it was a letter to the Veterinary Record that first associated the disease with CAstV and described the first Finnish case, which appeared in a single broiler-breeder flock in 2006, with greater numbers of flocks infected in subsequent years [4]. The progeny from this first flock was unusually pale, hence the condition was termed “White Chick”. In addition to Finland, white chick hatchery disease has been reported in several countries including Poland, Brazil, Canada and the US [5,6,7,8,9]. Anecdotal reports and diagnostic testing show a wider global prevalence, including the Middle East, Norway, Sweden and France. Losses due to reduced hatchability can be very high, with Finnish veterinarians reporting an average hatchability drop of 29% across affected flocks, with one individual farm experiencing a loss of 68% over a two-week period [4].

This paper aims to review the causative agent, clinical signs and disease progression, histopathology, economic impact, differential diagnosis, transmission, prevention and control and methods of detection and describe the genetic diversity of the virus strains that cause white chick hatchery disease. In addition, using protein modelling of the capsid spike proteins, it will attempt to explain possible reasons for the unique presentation of white chick hatchery disease.

## 2. Determination of the Causative Agent of White Chick Hatchery Disease

Particular strains of CAstV were purported to be the causative agent of white chick hatchery disease when it was initially isolated from three separate cases in Finland, Norway and Canada [4]. There are other pathogens capable of producing the clinical signs observed in white chick hatchery disease, such as runting, a decrease in egg production, a decrease in hatchability and an increase in dead in shell embryos; therefore, discriminatory testing of suspected birds was important to establish potential aetiology. In the reports on the Finnish cases of white chick hatchery disease, initial testing of chicks included the use of specific molecular assays for infectious bronchitis virus (IBV), infectious laryngotracheitis virus, avian influenza virus, Newcastle disease virus, paramyxovirus -2 and -3, fowl adenovirus (egg drop syndrome virus), avian rhinotracheitis virus, *Mycoplasma gallisepticum* and *M. synoviae*, and subsequent testing was for avian nephritis virus (ANV) and CAstV [4]. In addition to some of the pathogens listed above, avian reovirus, avian rotavirus, infectious bursal disease virus and chicken parvovirus were also excluded as the causative agents in the Brazilian and Polish white chick hatchery disease cases [7,8]. The results were negative for all pathogens tested except for CAstV, which was detected in multiple organs, including kidneys, liver [4,8], intestines and yolk sac [7]. Additionally, in the Finnish cases, feed contamination, toxin involvement and mineral deficiencies were not directly tested but were excluded as possible causes since other healthy flocks ate the same batch of feed, and the disease manifestations in affected birds cleared while eating the same batch of feed [4], further affirming CAstV as the probable causative agent of white chick hatchery disease.

Virus isolation was performed on chick samples affected by white chick disease in Finland, Norway and Canada in specific pathogen-free (SPF) embryonated eggs yielding three isolates. Typing of the CAstV isolates from these cases using the hypervariable capsid gene (open reading frame 2 (~2.2 kb)), which is often used for typing and phylogeny, displayed high levels (>90%) of shared nucleotide and amino acid identity and were typed as belonging to the B serogroup in a newly created subgroup, Biv. This finding suggested that CAstV might be the causative agent since typically many diverse CAstV strains are detected in circulation that can vary by more than 50% in the capsid gene; therefore, this high degree of strain similarity appeared significant. The CAstV isolates were inoculated into eggs, causing dwarfing and embryo death and partially fulfilling Koch’s Postulates [4]. Further molecular and serology testing of multiple white chick samples from Finland showed that 13 breeder flocks, which were the sources of affected eggs and chicks, had been mainly seronegative for B group CAstVs prior to the disease episode but were all seropositive afterwards, and CAstV was quantified at high levels (>6 logs) in affected chicks and dead-in-shell embryos [4,10].

In another experiment, involving a Polish CAstV isolate, Sajewicz-Krukowska et al. [8] inoculated the yolk sac of ten 8-day-old embryonated SPF eggs with allantoic fluids from infected eggs in which a Polish CAstV strain, PL/G059/2014, had been isolated from samples arising from cases of white chick hatchery disease. Of the ten infected embryos, one died on the day of inoculation and was recognised as a random, non-specific death. The remaining nine embryos were put in the hatchery, where they were allowed to hatch. A total of six embryos died during this experiment; one was very weak and died on the day of hatch, three started to pip but were too weak to emerge through the shell, and the remaining two embryos died in their shells [8]. Of the three embryos that hatched and survived, two appeared normal, while one was weak and runted with white plumage and died six days post-hatch. Pale, large livers covered in haemorrhages and lesions were observed during a post-mortem of the three chicks that failed to hatch, along with enlarged spleens, kidneys and pancreases and the presence of gelatinous oedemas around the head and neck of the chicks. This small study further demonstrated Koch’s Postulates by the replication of white chick hatchery disease using a CAstV isolate cultured from diseased samples that resulted in weak chicks that failed to hatch, dead-in-shell embryos and a small, weak chick with white plumage [8]. However, this strain is genetically very different from the previously described strains and was typed to a different CAstV serogroup, A, as opposed to the B serogroup, which contains the Finnish and other strains. To date, this is the only published strain associated with white chick disease that has not been typed in the B serogroup, but there is evidence of genomic recombination which may explain similarities in disease presentation and which is discussed in the later section on genetic diversity.

## 3. Clinical Signs and Disease Progression

Typically, there are no clinical signs of infection in laying breeder flocks, other than a possible transient egg drop in the order of 3–15 per cent [4,5]; however, this was only observed in some of the flocks so that increases in mid- and late-term embryonic deaths and poor hatchability may be the first indications of a white chick infection two to three weeks after the breeder flock became infected. In Finland, it was reported that adult hens typically became infected between 30 to 40 weeks of age [4] (Figure 1) whereas in Canada, affected flocks ranged from 27 to 57 weeks [5]. These naïve birds will not have experienced a previous infection by the CAstV strains associated with white chick disease, although infection by other CAstV B group strains may confer some protection. In Finnish flocks, the hatchability drops typically lasted for approximately two weeks until, presumably, the adult birds seroconverted, thereby preventing viral transmission and hatchability was restored [4] (Figure 1). However, in Poland, the disease was observed for up to four weeks, possibly due to the genetic differences observed between the Finnish and Polish CAstV strains or to the Polish flocks having a poorer hygiene status [4,8].

As demonstrated in Figure 1, the strains of CAstV that cause white chick hatchery disease can result in severe embryo mortality and culls of the runted, weak, white chicks produced during this 2–4 week period. A drop in hatchability is frequently observed in cases of white chick hatchery disease (Figure 1), with an average drop of 29% experienced in Finland; however, there were reports of individual cases as high as 68%. This trend is similar to other countries, including Canada [11], although the strain of CAstV that infected Polish flocks was reported to cause less mortality, with a decrease in hatchability of approximately 5% [8]. The decreases in hatchability indicate that the virus was vertically transmitted, which was confirmed by the detection of CAstV RNA in embryos that were found dead-in-shell by quantitative, real-time RT-PCR tests and in affected chicks that hatched with high CAstV viral loads [4]. Infected chicks that survive hatching are identified by their distinct white plumage and are typically weak and runted, compared to non-infected chicks. The lack of yellow pigmentation in the chicks’ feathers is possibly due to viral interference resulting in the reduced transfer of carotenoids that are circulating in the mother into the egg, or more probably, the presence of CAstV in the egg preventing embryonic absorption of carotenoids from the yolk. The mechanisms underlying white chick hatchery disease clinical signs are currently unknown.

## 4. Histopathology

In the paper from Smyth et al. [4], observations included hypertrophic bile ducts, inflammation of the heart and granulocytes in the bursa of Fabricius, kidney, liver tissue and heart of chicks with white chick hatchery disease in Finland. The liver was suggested as being one of the more commonly infected organs in affected chicks, with a green and mottled appearance. Observations in late dead embryos included clumps of granulocytes in liver tissue as well as areas of necrotic tissue containing inflamed blood vessels, resulting in vasculitis and enlarged bile ducts [4].

Sajewicz-Krukowska et al. [8] also identified that the liver in chicks and embryos infected by the Polish strain of CAstV had the most severe lesions and were also green in colour and mottled. The Brazilian strain of CAstV associated with white chick hatchery disease was primarily detected in the gizzards, intestines, lungs, kidneys, pancreas and spleen, as well as the yolk, but less frequently in the liver, heart, brain and the proventriculus in chicks affected by white chick hatchery disease [7]. Baxendale and Mebatsion [12] identified other strains of CAstV in the white blood cells of infected birds, and, although the CAstV strains associated with white chick hatchery disease have not specifically been reported in the blood of infected birds, it is highly probable that they also circulate in the blood, resulting in systemic infection.

## 5. Economic Impact

Figures for the economic impact of white chick hatchery disease are generally not available, but an in-depth Canadian study in 2017 revealed that white chick hatchery disease has infected Ontario poultry farms for 30 years [5]. In this Canadian study, the impact of twelve cases of white chick hatchery disease on two hatcheries in Ontario was estimated. Drops in broiler breeder egg production observed in flocks in Ontario were as high as 15% in some cases. This resulted in a considerable economic loss for the egg producer of up to approximately CAD 7000 per 10,000 birds in the most severely affected flocks. Decreases in the hatchability of eggs ranged from 2% to 49%, equating to a financial loss for paid eggs of approximately CAD 300 to CAD 10,500 per 10,000 eggs. The total combined economic loss for all of the affected egg producers in Ontario was approximately CAD 71,000 per 10,000 eggs [5]. Economic losses were also experienced by the hatchery company. Between June 2015 and November 2015, for adjusted flock size between two hatcheries that are owned by the one company, per 10,000 hens, approximately 106,000 chicks did not hatch, and a further 6000 chicks were lost because of culling due to the effects of white chick hatchery disease [5]. The total loss for the hatchery company for unrecovered fixed costs and chick sales margins when the flock size is adjusted to 10,000 hens was CAD 21,000. The combined impact for the egg producer and hatchery company during this five-month period was approximately CAD 92,000 per 10,000 hens and CAD 142,000 for unadjusted flock size [5]. As white chick hatchery disease appears to have similar effects on broiler chick production levels in other countries as with Canada, economic losses are expected to be similar to these losses in Canada.

## 6. Differential Diagnosis

The main clinical signs of white chick hatchery disease are an increase in mid-to-late embryo deaths and runted, weak chicks with pale down that do not survive for long after hatching. Despite *Mycoplasma* and Newcastle disease causing a reduction in egg production [13,14,15], and fowl adenovirus, reoviruses and *Salmonella* being vertically transmitted pathogens [16,17,18,19,20], these pathogens do not cause the main clinical signs of white chick hatchery disease and have not been detected in cases of white chick hatchery disease.

White chick hatchery disease appears distinct from pale bird syndrome, a malabsorption or enteric disease frequently reported in the 1970s and 1980s [21,22,23]. Pale bird syndrome is most commonly observed in 1–6-week-old broiler chicks, and typical clinical signs include weakness, poor feed conversion, undigested feed in the faeces, runting and lack of pigmentation of the feathers, eggs, legs and beaks [23,24,25,26]. Causative agents associated with pale bird syndrome are mycotoxins, including aflatoxin [27], coccidial infections [28] and reoviruses [29,30] and are associated with the malabsorption of nutrients such as carotenoids in the gastrointestinal tract, which are responsible for yellow pigmentation in the birds’ down and egg yolk [22,24,25,27,31]. To our knowledge, there are no reports associating astroviruses with pale bird syndrome cases; (however, it is important to note that CAstV was only recognised as a separate species in 2004), nor are there reports of cases in day-old chicks, reductions in egg-laying or reductions in hatchability in pale bird syndrome. Therefore, despite some similarities in clinical signs, white chick hatchery disease and pale bird syndrome should be recognised as two separate diseases.

A similar enteric disease to pale bird syndrome is helicopter wing disease, associated with retarded feather development, reduced growth and less frequently, diarrhoea, lymphoid and pancreatic atrophy and an increase in mortality [32]. Mettifogo et al. (2014) state that these diseases are described as enteric diseases due to their clinical manifestations, which are caused by the same pathological agents, some of which may be vertically transmitted, resulting in hatchery diseases.

## 7. Transmission

Multiple highly variable strains of CAstV have been detected in broiler houses, since it appears to be one of the most common enteric viruses in broiler flocks globally, resulting in the horizontal transmission of the virus, most commonly via the faecal–oral route [12,33,34,35]. The virus is excreted in the faeces, which can also contaminate the water systems of poultry houses, and is consumed by non-infected birds [36]. However, due to chicks shedding high levels of CAstV upon hatching and the detection of CAstV in dead-in-shell embryos, it is thought that the CAstV strains that cause white chick hatchery disease are transmitted by vertical transmission [4,5,7,8]. Evidence to support the vertical transmission of CAstV has been largely inferred as no studies have as yet confirmed the exact mechanism of transmission; however, in the case of thirteen Finnish breeder flocks, the hens were mainly seronegative before the white chick hatchery disease episode while all had seroconverted shortly after, strongly suggestive of vertical transmission [10]. In addition, CAstV was detected in dead embryos and at high levels in just-hatched chicks [4].

It is thought that the virus passes from the parent into the egg before the parent develops immunity to the virus. However, the mode of transmission has not been conclusively demonstrated, as to whether transmission occurs via the reproductive organs during the development of the egg, during fertilisation between the egg and semen or via pseudo-vertical transmission, where the virus is absorbed through the eggshell and infects the embryo. It is speculated that the more probable form of CAstV transmission occurs when the immature egg is within the hen’s reproductive tract in order to result in an egg drop, which is reported in some cases [4]. This will result in the virus being transmitted into the yolk sac so that the virus may enter the embryo, infecting the gut and other tissues and thereby impairing its growth, causing runting, chick weakness and impacting hatchability rates. If the virus was absorbed into the egg from faeces while being laid, as in pseudo-vertical transmission, it would not result in a decrease in the numbers of eggs laid, although a decrease in the numbers of eggs hatching would be possible.

## 8. Prevention, Control and Treatment

Current control methods for white chick hatchery disease include attempts to reduce the occurrence of CAstV infections by developing and maintaining good flock biosecurity. However, white chick hatchery disease is regarded as a clean barn disease and, therefore, high levels of biosecurity may leave breeder flocks susceptible to subclinical infections by these pathogenic strains of CAstV during lay if they have not developed specific immunity during rear. In the absence of a CAstV vaccine, and since it has been observed that antibodies to heterologous B serogroup strains confer protection against other B serogroup strains (unpublished results), it may be preferable for breeder flocks to seroconvert against circulating CAstV B serogroup strains during rear, rather than experience vertical transmission of pathogenic strains during lay.

Since CAstV is relatively resilient to the disinfectants typically employed by farmers compared to other pathogens, complete sanitation of poultry houses is recommended, including walls, feeders, fans and water systems, to effectively remove the virus [36]. Sufficient downtime is recommended between breeder flock depopulation and repopulation with birds close to the start of the laying period after a robust cleaning and disinfection regime. The use of 10% bleach as a disinfectant was not found to be successful at fully removing the virus from poultry houses. Instead, Koci and Schultz-Cherry [36] recommend using 90% methanol, 1.5% virkon or 0.3% formaldehyde, which appear to be required for successful disinfection. Removal of the darkling beetles, which are known to harbour CAstV infections in their carapaces and bodies, is also recommended [37]. Fogging houses and other affected locations, such as hatcheries, after cleaning is recommended as the most effective means to inactivate residual CAstV, where naïve chicks may be at risk of horizontal transmission. Strict biosecurity measures should be employed: farmers, veterinarians and feed lorries should schedule arrival to different farms carefully, visiting non-infected farms first, followed by infected farms to prevent the spread of infection. Litter should be disinfected, removed and changed between each flock rotation and disposed of in a way that prevents runoff and re-contaminating the houses [36].

Veterinarians have reported that breeder flocks experience white chick hatchery disease only once during their life cycle [4], which implies that infected adult birds develop lifelong immunity to this strain of CAstV. This suggests that a breeder vaccine based on a suitable CAstV serogroup B strain and, in light of the Polish white chick hatchery disease findings, possibly also a CAstV serogroup A strain, might be able to provide adequate flock protection, enabling birds to build a long-term immune response which will protect against the disease. This would prevent vertical transmission and therefore avoid the severe economic consequences of white chick hatchery disease. In addition, such a vaccine should also protect against other hatchery diseases caused by different CAstV B groups strains and would provide maternal antibodies to potentially protect offspring against horizontally acquired CAstV infections [2,38], such as the CAstV subgroup Biii strain that causes kidney disease with visceral gout in the neonatal period, when they are highly susceptible.

## 9. Methods of Detection

Originally, the detection of astrovirus infection was performed by electron microscopy (EM) to identify the characteristic five/six-point star shape of the virion [39]. It is this star shape that is described by the Greek word “Astron,” from which the virus name derived. However, this star shape is pH-dependent and present in less than 10% of EM preparations. Furthermore, virus isolation of astroviruses is slow, while some strains are difficult to culture in cells and identification of isolates relies on skilled operators of electron microscopy. Previously, avian astroviruses were misclassified as picornavirus-like or enterovirus-like [40] due to similar characteristics; therefore, molecular testing is now routinely applied [12].

Application of the real time, quantitative, pan-CAstV reverse transcription-polymerase chain reaction test (RT-qPCR) [41] can be used to quantify CAstV in samples from chicks. A conventional PCR can be performed on the capsid gene if sufficient CAstV (ideally ≥ 10^6^ genome copies) is present in the samples. The PCR amplicon may be typed by DNA sequencing to determine in which CAstV subgroup the strain belongs via a multiple alignment with other CAstV capsid gene sequences, including those from other white chick hatchery disease cases. For convenience and cost minimisation, partial capsid gene sequences have often been used for typing; however, as a recent paper from Canada has shown, CAstV genomic recombination may confound typing based on partial genomic sequence. Therefore, the entire short genome of ~7 kb should ideally be sequenced and compared [42], which could readily be achieved using next-generation sequencing (NGS) technologies. However, it remains to be seen whether genomic typing provides a more academic value than practical benefit.

The parent/breeder flocks can be sampled for seroconversion against CAstV (B group) using an enzyme-linked immunosorbent assay (ELISA) [38] to confirm that an infection has occurred in the parent flocks, ideally comparing sera from before and after the event. Although it has not been established definitively, it appears that breeder flocks naïve of any B group infections are most susceptible to white chick hatchery disease infections and blood may be seronegative prior to the white chick episode. Flocks that have been infected by other B group strains may have some protection against infections by the strains causing white chick hatchery disease as there is serological cross-reactivity across the B subgroups. However, since this ELISA does not discriminate between antibodies raised against different B subgroup infections, caution must be applied in the interpretation of the results; however, it would be reasonable to expect that recently infected breeder birds would display higher titre groups with smaller coefficients of variation than flocks infected previously by other B group strains. Nor will it detect antibodies to CAstV A group infections and so would not detect antibodies raised by the Polish white chick CAstV strain (Aiii). By the time of hatching, affected parent flocks may have cleared the infection, and so breeder tissues may test negative by RT-qPCR; therefore, serological testing is the best means for establishing that a positive infection has occurred.

## 10. Classification of CAstV

CAstV is an avian astrovirus of the genus *Avastrovirus* and the *Astroviridae* family. Astroviruses are associated with diarrhoea, enteritis and growth problems in the young of many different animal species, including wild birds, ducks, chickens and most mammals. Astroviruses are small, round, non-enveloped, positive-sense RNA viruses with a single-stranded genome. Avian astroviruses have historically been named according to their host, for example, duck astrovirus. However, since astroviruses can cross species barriers and infect other species, this classification system is being replaced by the International Committee on Taxonomy of Viruses, which classifies astroviruses based on the amino acid sequence of the capsid protein [6]. Among the avastrovirus genus, three species are recognised, *Avastrovirus 1, Avastrovirus 2 and Avastrovirus 3*, with CAstV being classified as Avastrovirus 2. In 2011, when this replacement classification was created, CAstV capsid gene (ORF2) sequences were not widely available; therefore, CAstV was referred to as “related viruses which may be members of the *Avastrovirus* genus but have not been approved as species” [6,43]. It should also be noted that, while ANV is also an astrovirus of chickens, ANV and CAstV are different species of the virus with distinct antigenicities and that this review is concerned only with CAstV.

## 11. Genomic Organisation of CAstV

The genome of CAstV, including those strains that cause white chick hatchery disease, is approximately 7.5 kbp in length and possesses a similar genomic organisation to other astroviruses. The CAstV genome is comprised of a 5′ untranslated region (UTR); three open reading frames (ORFs 1a, 1b and 2), which code for a non-structural polyprotein (including the serine 3C protease, which cleaves the polyprotein into functional peptides) and a potential VPg in ORF 1a [44]; an RNA-dependent RNA polymerase in ORF 1b and the viral capsid in ORF 2; and a 3′ UTR and a polyadenylated tail [43,45]. A frameshift structure separates ORF1a and ORF1b, enabling the translation of the two ORFs directly from the positive-sensed genomic strand, forming a polyprotein which is subsequently cleaved into two subunits, while ORF2 is expressed from sub-genomic messenger RNA [36,45,46,47].

## 12. Genetic Diversity of CAstV

Currently, most molecular characterisation of different CAstV strains is based on the nucleotide and/or amino acid sequence of the capsid precursor of ORF2 since this is the most hypervariable region between CAstV genomes [48]. Due to this variation in capsid sequence, CAstV is divided into two distinct serogroups, A and B, which share <40% of amino acid homologies over the entire capsid gene [49] and low levels of serological cross-reactivity [2]. Initially, group A was divided into three sub-groups, which are genogroups based on nucleic acid clustering rather than serology, and B into two sub-groups with 77–82% inter-subgroup homologies and >85% intra-subgroup homologies within each group, respectively [49]. However, two new B subgroups were added, one containing strains associated with white chick hatchery disease (Biv) [4,42] and the other containing strains associated with kidney disease and visceral gout (Biii) [50] (Figure 2). As shown in Table 1, further subgroups are probable based on the wide genomic diversity of CAstV. While groups A and B are serologically distinct, there is cross-reactivity within the B subgroups and probably also between the A subgroups; however, the latter is yet to be empirically determined [38].

Commonly, CAstV strains that cause white chick hatchery disease cluster most closely within the subgroup Biv, based on ORF 2 sequences, including several strains from Canada and Brazil (Figure 2). The Biv subgroup is highly conserved as its members share 95% to >99% amino acid homologies. Palomino-Tapia et al. [42] identified 14 novel CAstV strains as the causative agents of white chick hatchery disease in recent outbreaks in Western Canada. Phylogenetic analysis of the entire ORF2 capsid proteins revealed that these strains are assigned to sub-group Biv of CAstV, where they cluster with the American CkP5 and CC_CkAstV CAstV strains that are associated with runting stunting syndrome (RSS) [55] (Figure 2). From an in-depth analysis of these strains, Palomino-Tapia et al. [42] identified that a total of 12 novel recombination events have occurred between CAstV strains belonging to the B serogroup, giving rise to some of these novel strains of CAstV. Furthermore, they propose that analysis of the entire CAstV genome is essential to correctly assign strains to the appropriate clusters. They illustrated the potential to misclassify strains using partial ORF2 sequence data, citing the example of Long et al. [11], who performed phylogenetic analysis on the first 644 nucleotides of ORF2 from CAstV strains isolated from white chick hatchery disease cases in Ontario in 2017 and 2018, classifying them in the CAstV Bii cluster. Analysing the same 5′, 644 nucleotide region of ORF2 of the 14 Biv strains identified recently from white chick cases resulted in the misclassification of these strains as Bii, and so the authors suggested that the Ontario strains may also belong to the Biv subgroup. However, phylogenetic analysis will need to explore the entire capsid sequence to accurately determine this [42].

In contrast to the Canadian strains, a Polish isolate from a case of white chick hatchery disease (accession number KT886453) was assigned to the distant subgroup Aiii (Figure 2) by sequence analysis of ORF2 [43]. Although chicks experienced a decrease in hatchability and runting when infected with the Polish strain, there were fewer chicks with white plumage and a less severe reduction in hatchability experienced than with other strains [8]. At the genome level, this Polish strain shares ~74% nucleic acid homologies with the Canadian Biv strains in Figure 2 but has only ~50% nucleotide sequence homology to the Canadian Biv strains in ORF2, demonstrating the higher levels of diversity within ORF 2. Sajewicz-Krukowska and Domanska-Blicharz [43] suggest that the Polish Aiii strain may be a recombinant due to its topology and to its genomic regions containing nucleotide similarities with different CAstV strains. A preliminary examination of the whole genome sequence of the Polish strain kindly performed by Dr. Victor Palomino-Tapia using Simplot [56], RDP5.5 [57] and Bootscan [58] programmes suggests that the Polish strain is a recombinant composed of a Bii-like major parent with a Biv-like minor parent across ORFs 1a and 1b, but did not detect recombination events in ORF2 (personal communication). However, the limited availability of CAstV genome and ORF 2 sequences, especially those of Group A, in the international databases may have restricted the scope of this recombination analysis. The presence of recombinant regions from Group B strains in ORFs 1a and 1b of the Aiii Polish strain may help explain some of the similarities in disease presentation and tropism, while the very distant Group A ORF 2 possibly accounts for the lower level of pathogenicity, which is thought to be largely determined by ORF 2. It also supports the contention that accurate classification of CAstV requires a whole genome sequence [42].

ORF2 of CAstV encodes the capsid surface spike domain, which is suspected to be the major determinant of pathogenicity, tropism and immunogenicity. The sequence within ORF2 that codes for the CAstV capsid surface spike domains, which are thought to be involved in host cell attachment [59], was identified for each representative strain from each of the four B subgroups (Figure 3). Comparative protein modelling illustrates the structural differences in the capsid surface spike domains of the CAstV Biv subgroup in comparison to the B subgroups (Figure 4) and the putative avian cell receptor binding sites, annotated as sites A and B in Figure 3 and Figure 5.

## 13. Summary and Conclusions

This paper reviews the current knowledge about white chick hatchery disease. It is an economic burden to companies in many regions of the world. Despite there being multiple varying strains of CAstV circulating in the farming environment, the distinctive strains that largely belong to the Biv subgroup have been identified as the probable causative agent for white chick hatchery disease. Despite the adult birds becoming infected with white chick hatchery disease-associated CAstV strains, there are no clinical signs in the adult birds; however, there is a much higher mortality rate in their offspring, with typical observations including an increase in mid-to-late embryo deaths and small, weak chicks with white plumage.

Molecular testing has identified CAstV present in embryos as well as in chicks on the day of hatch, indicating probable vertical transmission of CAstV from the adult bird into the embryo. In field observations, it has been determined that when naïve, in-lay breeder flocks experience white chick hatchery disease, it takes approximately two weeks for hatchability to restore to expected levels, which is typical of a vertically transmitted pathogen. During this two-week period, it is hypothesised that CAstV is transmitted into the eggs/embryos, and white chick hatchery disease is subsequently observed in the progeny chicks. After this two-week period, adult seroconversion appears effective at preventing any infections in subsequent progeny. A CAstV Group B breeder vaccine, based on a suitable strain(s), could be highly effective at preventing vertical transmission of CAstV B group strains, including the Biv strains associated with white chick hatchery disease and protecting neonates against environmental (horizontal) CAstV infections.

Recombination analysis has identified that many of these Biv strains and the anomalous white chick Polish Aiii strain have evolved due to recombination events between various diverse CastV strains, giving rise to the emerging white chick hatchery disease. Further analysis of the amino acid sequences that code for the capsid surface spike domains and putative avian receptor-binding sites within the CAstV Biv subgroup strains have demonstrated the genetic diversity of these strains compared with other B-group CAstV strains, which we speculate could potentially be attributed to the differences in CAstV-associated diseases. The proposed capsid surface spike domain amino acid sequence and comparative protein models illustrated in this paper provide information on the structure of the spike domain and could aid in the development of a pan-Group B CAstV vaccine, which may prove effective at preventing hatchery disease caused by strains of CAstV in other Group B subgroups. Since the current methods of biosecurity are not 100% effective in preventing infections in flocks, a vaccine administered to breeder hens would be an effective control method to prevent the widespread loss of chicks worldwide and the detrimental financial losses to farmers and poultry companies.

## Figures and Tables

**Figure 1 viruses-13-02435-f001:**
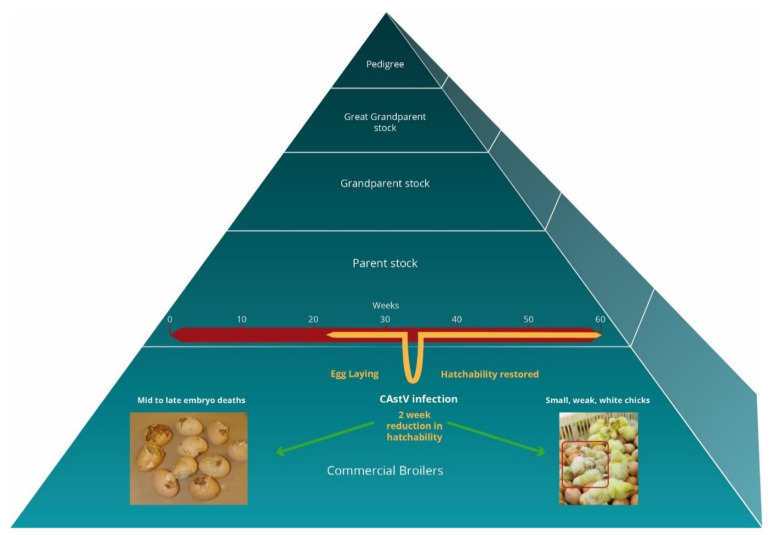
A schematic diagram of the poultry breeding pyramid illustrating the typical timeline of white chick hatchery disease during parent stock egg-laying (yellow line). The dip in the yellow line indicates a typical CAstV infection and subsequent hatchery drop with vertical transmission of virus, resulting in white chick hatchery disease in broiler progeny prior to parent flock seroconversion against CAstV and consequent recovery of hatch levels.

**Figure 2 viruses-13-02435-f002:**
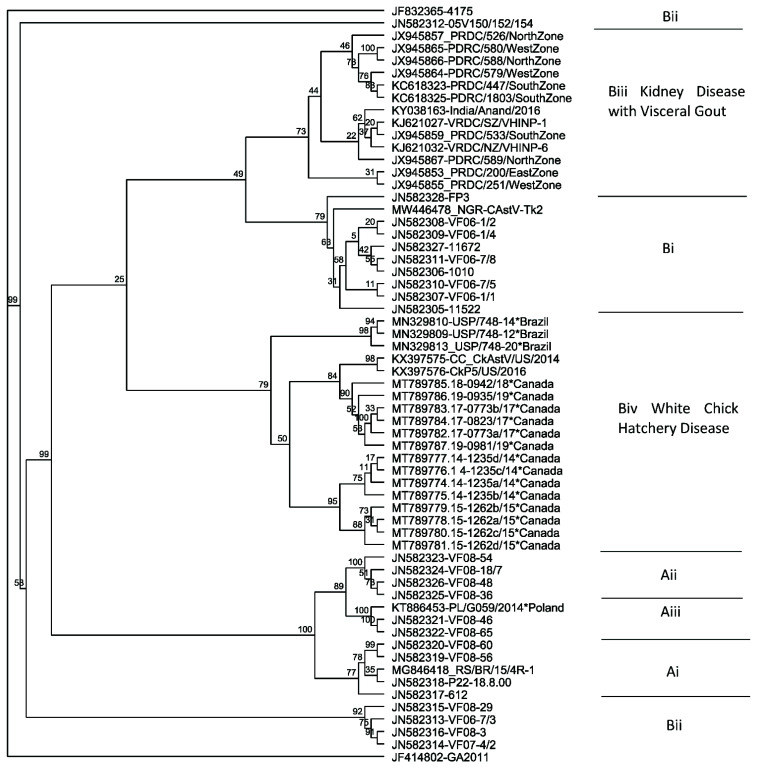
Phylogenetic tree of CAstV complete ORF 2 amino acid sequences. Sequences clustered according to sub(geno)groupings of the A (subgroups Ai, Aii and Aiii) and B (subgroups Bi, Bii, Biii and Biv) groups, which are serologically distinct. The maximum likelihood tree was constructed in RAxML version 8 [51] on Geneious version 2021.1 (Biomatters) using the GAMMA BLOSUM62 protein model with 1000 rapid bootstrapping replicates and searching for best-scoring ML tree with 456 parsimony random seed.

**Figure 3 viruses-13-02435-f003:**
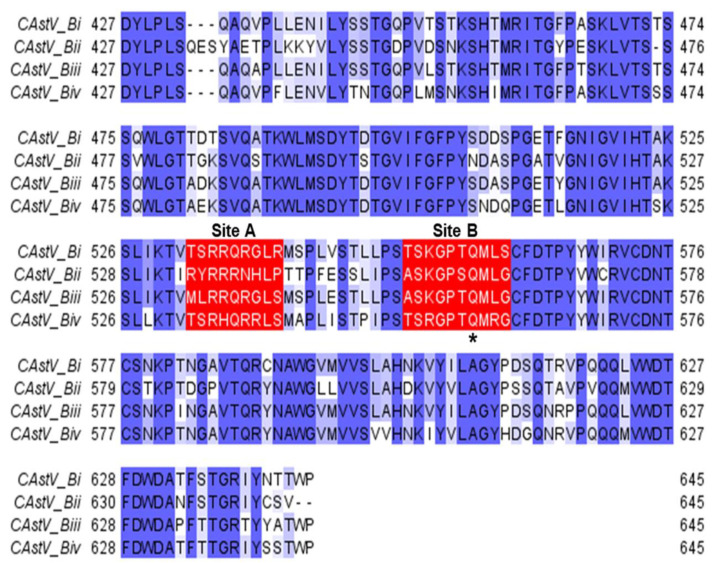
Multiple sequence alignment of representative B-group CAstV capsid protein surface spike domains. The Bi-group surface spike domain sequence was from strain JN582327_11672, the Bii-group from strain JN582316_VF08-3, the Biii-group from strain KC618325_PDRC/1803/South and the B iv-group from strain MT789787_19-0981/19_Canada *. Conserved, similar and non-conserved amino acids are highlighted in dark blue, light blue and white, respectively. Two clusters of amino acid residues that are candidates to form putative avian cell receptor-binding sites (annotated as ‘Site A’ and ‘Site B’) of the spike domain are boxed in red. Site A consists of residues 532–540 (532–542 for the Bii-group CAstV) and site B consists of residues 552–561 (554–563 for the B ii-group CAstV). Site B contains a highly conserved glutamine residue, indicated by an asterisk, that is the structural equivalent to Q552 of the turkey TAstV-2 capsid protein spike domain, and deletion of which results in increased incidence of disease in turkeys infected with TAstV-2 [59,60]. Alignments were performed with the online ClustalW server [61].

**Figure 4 viruses-13-02435-f004:**
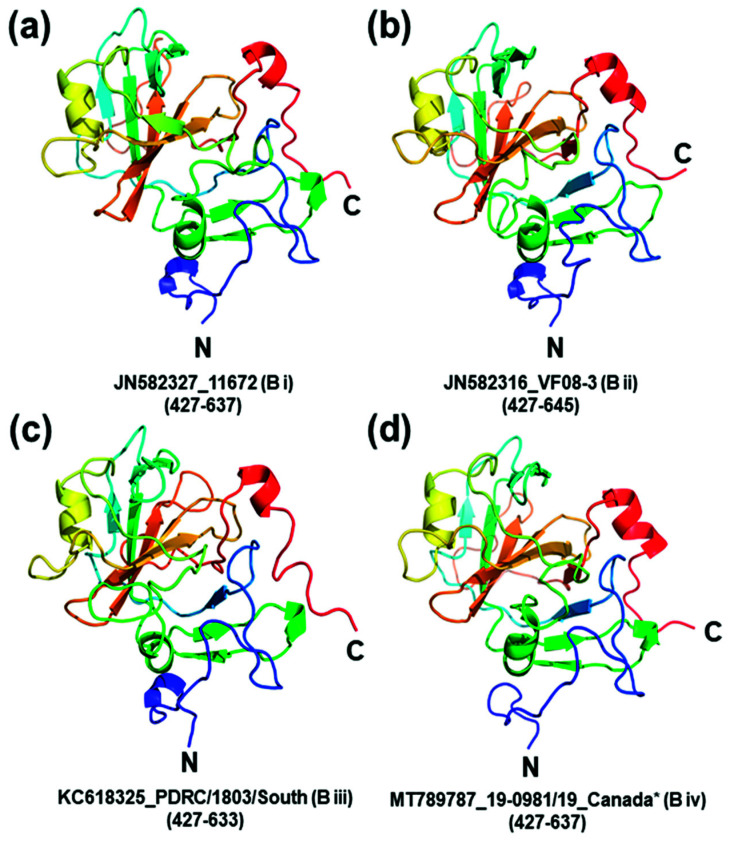
Comparative protein modelling reveals structural differences between CAstV capsid surface spike domains of the B-groups of the virus. Each cartoon model represents a side view of one dimer subunit of the CAstV capsid surface spike domain with the N-terminal located at the bottom and the homodimer interface on the right-hand side. (**a**) Bi-type spike domain from strain JN582327_11672; (**b**) Bii-type spike domain from strain JN582316_VF08-3; (**c**) Biii-type spike domain from strain KC618325_PDRC/1803/South; and (**d**) Biv-type spike domain from strain MT789787_19-0981/19_Canada*. The residues modelled are indicated in parentheses. Although the modelled proteins share the same general α/β structure, consisting of a scaffold that comprises an antiparallel β-barrel with a tightly packed hydrophobic core, differences that define each of the B-type spike domains are apparent. The homology models were built and refined with MODELLER [62] using the 1.5-Å resolution crystal structure of turkey astrovirus 2 (TAstV-2) capsid surface spike domain (PDB ID: 3TS3) as the structural template [59]. The models were visualised using the PyMOL Molecular Graphics System [63].

**Figure 5 viruses-13-02435-f005:**
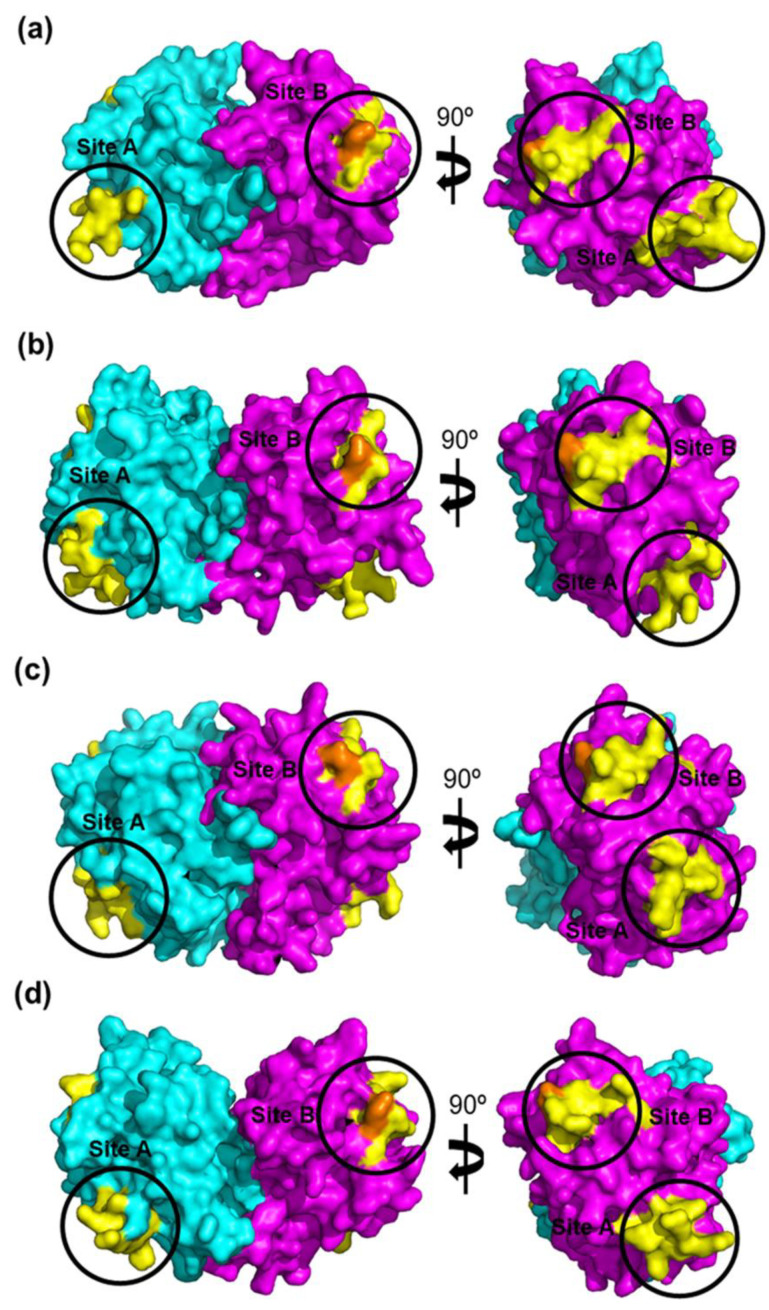
Identification of putative avian receptor-binding sites in B-group CAstV capsid protein spike domains. Structural models of the CAstV B-group spike domains are shown in the biologically relevant homodimer form and depicted in side-view as surface representations. The dimers were built by mapping the homology model monomers described in Figure 4 onto the coordinates of the TAstV-2 spike domain homodimer [59]. One dimer subunit is coloured cyan and the other is coloured magenta. Amino acid residues suggested by sequence alignment (Figure 3) to form potential avian cell receptor-binding sites are coloured yellow. These form two distinct surface-exposed clusters circled in black and annotated as Site A and Site B, located diagonally opposite one another on the top and bottom edges of each dimer subunit. In each model, a highly conserved glutamine residue (located in Site B) that is the structural equivalent of Q552 of the turkey astrovirus TAstV-2 spike domain is coloured orange. We speculate that amino acid differences between the putative receptor-binding site regions in the different B-group viruses could form the molecular basis for receptor recognition and binding and could also explain differences in disease incidence of the B-group CAstVs. It is intriguing to further speculate that each B-group CAstV may recognise and bind at least one different cell receptor. (**a**) Bi–group spike domain homodimer from strain JN582327_11672; (**b**) Bii–group spike domain homodimer from strain JN582316_VF08-3; (**c**) Biii–group spike domain homodimer from strain KC618325_PDRC/1803/South; and (**d**) Biv–group spike domain homodimer from strain MT789787_19-0981/19_Canada*. The models were visualised using the PyMOL Molecular Graphics System [63].

**Table 1 viruses-13-02435-t001:** Diseases associated with CAstV subgroups.

CAstV Subgroup	Strain	Accession Number	Associated Disease	Clinical Signs and/or Experimental Findings	Organs Infected	References
Ai	612	EU669001	Originally isolated from broilers suffering from respiratory distress	Growth retardation	Duodenum, jejunum and kidney. Isolated from birds suffering from respiratory distress	[40,52]
Aii	VF08-54	JN582323	N/A	Underperformance	Intestines	[49]
	VF08-36	JN582325				
Aiii	VF08-46	JN582321	N/A	Underperformance	Intestines	[49]
	PL/G059/2014	KR052479	White chick hatchery disease	Adult birds—no apparent effect Chicks—runted, weak chicks, white plumage, decreased life expectancy and a decrease in egg hatchability	Liver, kidneys, pancreas, spleen	[8] Sajewicz-Krukowska et al., 2016
Bi	NGR_CAstV_Ch1	MK509014	Hatchery disease	Growth retardation, weakness, dullness, ruffled/wet feathers, splayed legs and hatchability problems	Intestines	[35]
	FP3	JN582328	Hatchery disease	Dead-in-shell embryos and weak chicks	Small intestine, kidney, pancreas	[53]
Bii	VF08-29	JN582315	N/A	Underperformance	Intestines	[49]
	GA2011	JF414802	Runting stunting syndrome (RSS)	Ruffled feathers, growth retardation and diarrhoea	Small intestine	[54]
Biii	PDRC 588	JX945866	Visceral gout and kidney disease	Embryo stunting, swollen, pale kidneys, liver necrosis, leading to kidney disease and visceral gout	Kidneys and liver	[50]
	PDRC 1804	KC618324				
	PDRC 579	JX945864				
Biv	16-028568-0005 ^a^	KY635984 ^a^	White chick hatchery disease	Adult birds—The health status of adults is normal other than occasional reduction in egg-laying. Birds test positive for antibodies against the CAstV B group Chicks and embryos—runted, weak chicks, white plumage, decreased life expectancy, decrease in hatchability, yolk sac remnant, oedema on the neck and head, subcutaneous oedema and enlarged, green, mottled livers	Adults—Cloacal cells Chicks—Liver, kidneys, intestines, pancreas, bursa of Fabricius spleen, yolk, gizzards, heart, brain, lungs and proventriculus Embryos—liver, kidneys, intestines and yolk	[11]
	18-0942/18	MT789785				[42]
	USP541-15 ^a,b^	KR013249 ^a,b^		Additional findings in chicks—Pale kidneys, pancreas, beaks and legs		[7]

^a^ denotes CAstV strains that are not included in the phylogenetic tree in Figure 2 as the amino acid sequence available in GenBank is not of the entire ORF2. ^b^ indicates strains available on GenBank which have not yet been published.

## Data Availability

All nucleotide sequences are available to download from GenBank.

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
