# Peer review of "A Review of the Emerging White Chick Hatchery Disease"

_viruses, 2021, doi:10.3390/v13122435_

Round 1

Reviewer 1 Report

The topics of this review in very interesting for the scientific community indeed enteric disease and the white chick disease are a current problem and more data to explain the evolution and epidemiology are necessary

Moreovere this article analyzed the aa sequences and proposed. A capsid surface spike domain  protein model that can result in important support for the future CAstV B vaccine.

Indeed I  suggest to add to the title a reference at the aa sequence analysis and protein model 

Line 61-121  This part need a more elaboration of the information currently known, to evoide a list of report, and produc a clear global report on CAstV.

Line 209-223  Other. virus and  bacteria must be consider in the differential diagnosis

Line 220-223  the affirmation in these line need more attention  and more study are necessary to evaluate the pale birds syndrome as describe in the article reported below: 

ScientificWorldJournal. 2014; Emergence of Enteric Viruses in Production Chickens Is a Concern for Avian Health Elena Mettifogo, 1 Luis F. N. Nuñez, 1 Jorge L. Chacón, 1 Silvana H. Santander Parra, 1 Claudete S. Astolfi-Ferreira, 1 José A. Jerez, 2 Richard C. Jones, 3 and Antonio J. Piantino Ferreira 1 ,* :

“In the past, enteric disease has been called the pale bird syndrome and helicopter wing disease and was characterized by poor growth and retarded feather development. These symptoms are observed consistently along with the other less frequent clinical signs including diarrhea, increased mortality, and pancreatic and lymphoid atrophy [615]. Enteric diseases seem to be the most acceptable name for this clinical manifestation because it most appropriately reflects the consistency of the clinical findings and indicates that these cases are probably caused by the same infectious agents.2

Line 318-326  the method describe here is a common molecular biology approach so you can indicate the principal gene amplify for CAstV and avoid the list of operations

Line 329-331 to date the complete genome analysis is not a usefully method for diagnosis but for phylogenitic analysis and study. 

Line 334-336 The CastV are normally presente in the environmental, consequently the animal will became positive at the Ab during the life so usage of Ab as a diagnostic method need more attention in the diagnostic approach 

Line 235-236.  Is not clear the actual classification in 2021 of CAstV 

Figure 2 Insert in the legend also the definition of Bi Bii Biii Ai Aia Ai

Author Response

Please see the attachment.    Many thanks for your review.

Reviewer 2 Report

The article is well written and shows a conpendium of useful and updated information on the exposed subject.

Author Response

There were no revisions requested by this reviewer.  Many thanks for your time.